# Degradation of Paracetamol in Distilled and Drinking Water via Ag/ZnO Photocatalysis under UV and Natural Sunlight

**Dobrina Ivanova, George Tzvetkov and Nina Kaneva \***

Department of General and Inorganic Chemistry, Faculty of Chemistry and Pharmacy, University of Sofia,
1 James Bourchier Blvd., Sofia 1164, Bulgaria; dobrina.k.ivanova@gmail.com (D.I.);
george.tzvetkov@gmail.com (G.T.)
\* Correspondence: nina_k@abv.bg

**Abstract:** The present study demonstrates the synthesis and application of Ag/ZnO powder films (thickness of 4 μm) as photocatalysts for natural sunlight and ultraviolet (UV, 315–400 nm) irradiation. The synthesis procedure is simple and eco-friendly, based on the photo-fixation of silver ions onto commercial ZnO powder via UV illumination for the first time. The photocatalytic efficiency of the newly developed films is evaluated through degradation of paracetamol in distilled and drinking water. Our experimental evidences show that the Ag/ZnO nanostructure films are more active than pristine ZnO films in the photodegradation process. Namely, the photocatalytic efficiency of the films modified with $10^{-2}$ M concentration of silver ions achieve the highest degradation (D) percentages for paracetamol in both types of water ($D_{distilled} = 80.97\%$, $D_{drinking} = 82.5\%$) under natural sunlight. Under UV exposure, the degradation percentages are slightly lower but still higher than those achieved by pure ZnO films ($D_{distilled} = 53.13\%$, $D_{drinking} = 61.87\%$). It is found that the photocatalytic activity grows in direct proportion to the concentration of $Ag^+$ ions: ZnO < Ag $10^{-4}$/ZnO < Ag $10^{-3}$/ZnO < Ag $10^{-2}$/ZnO. Scanning electron microscopy, X-ray diffraction, X-ray photoelectron spectroscopy, UV–vis diffuse reflectance and photoluminescence spectroscopy are used to characterize the as-prepared ZnO and Ag/ZnO nanostructures. The improved photocatalytic performance of the Ag/ZnO films is mostly attributed to the combination of excited electron transfer from ZnO to Ag and the inhibition of photogenerated electron–hole pair recombination. Furthermore, Ag/ZnO nanostructure films can retain their photocatalytic activity after three cycles of use, highlighting their potential practical application for the treatment of pharmaceutical wastewater in real-world scenarios where natural sunlight is often more readily available than artificial UV light.

**Keywords:** heterogeneous photocatalysis; Ag/ZnO powder films; distilled and drinking water; UV light; natural sunlight; paracetamol

## 1. Introduction

Various pharmaceutical pollutants have been detected in wastewater [1,2], surface and groundwater [3,4] and even in drinking water [5,6]. More specifically, antibiotics, analgesics, antipyretics and hormones are frequently found in the aquatic environment [2,7]. As long-term consumption of pharmaceutical contaminants through drinking water results in carcinogenetic disorders that have a negative impact on the human health, the presence of these compounds in drinking water is a significant problem and causes public concern [6]. The harmful effects of pharmaceuticals include toxicity, resistance to pathogenic microorganisms, genotoxicity and endocrine disruption [8,9].

One of the most widely used medications—paracetamol (acetaminophen)—is a common, non-biodegradable and highly water-soluble compound that is present in more than a hundred pharmaceutical products [10]. Paracetamol (PCA) is frequently used to treat headaches, a variety of colds and influenza. According to a report from 2000, paracetamol ranks third among the top medications, and its consumption exceeds 400 tons

annually [11]. This analgesic has been discovered in natural and European wastewaters in amounts up to 10 g/L and 6 g/L, respectively [12]. Recently, many methods, including electrochemical [13,14], ozonation [15,16], Fenton and photo-Fenton [17,18] and $H_2O_2$/UV oxidation techniques [19,20], to mineralize PCA have been reported. However, one of the most promising AOPs (advanced oxidation processes) for the removal of PCA from water medium is heterogeneous photocatalysis.

Heterogeneous photocatalysis with semiconductor materials such as zinc oxide (ZnO) has indeed gained popularity as an effective method for treating wastewater contaminated with dyes, pesticides and pharmaceuticals [10]. ZnO is a low-cost, non-toxic, chemically stable and easy-to-produce material with a range of applications, including micro-gas sensing devices [19], solar cells [20], nonlinear optics [21], integrated photonic devices [22] and photocatalysts for wastewater treatment [23,24]. However, under visible or sunlight irradiation, ZnO cannot be used as a catalyst due to its significant band gap energy. Additionally, due to the rapid recombination rate of charge ($e^-/h^+$) couples, its photocatalytic efficiency is also constrained. To overcome this problem, the ZnO surface can be modified and functionalized with co-catalysts, typically noble metals, such as Pt, Pd, Ag and Au, and based on the formation of heterojunctions to limit the possibilities for recombination [25–27]. Different approaches have been used to combine the ZnO substrate with metallic co-catalysts [25–27]. Among them, the direct photo-fixation or photocatalytic deposition of metal clusters under UV light has gained attention recently [28,29]. More precisely, under the influence of the photogenerated charges on the ZnO substrate, the co-catalyst can be deposited as a result of the reduction in metal ions, which are typically from an aqueous solution [28,29]. As photogenerated electron acceptors, the noble metal ions ($Ag^+$, $Pt^{2+}$, $Pd^{2+}$, $Au^{3+}$) can be successfully reduced to the corresponding metals [30–32].

Most of the studies that have examined ZnO and co-catalytically modified ZnO focus on the systems using suspended photocatalyst particles. This limits their practical applicability due to the requirement of centrifugation or filtration to reuse fine ZnO particles. The design of systems where the photocatalysts are supported on inert substrates as layers and thin films proved to be successful in overcoming such obstacles [30]. Very recently, Hao and co-workers [33] described the construction of hydrothermally grown ZnO films on wire mesh subsequently decorated with Ag nanospheres using the impregnating photoreduction treatment. The formed Ag–ZnO heterojunction improves the absorption of UV and visible light, thereby boosting the photocatalytic properties of the films. Rati and co-workers [34] reported that Ag addition contributes to the improvement of ZnO thin films throughout the photocatalytic process of methylene blue degradation. Therefore, developing novel ZnO-based thin-film photocatalysts for water remediation is of constant interest.

The main goal of the present work is to elaborate a new fabrication strategy to grow Ag-modified ZnO films for photocatalytic removal of paracetamol. In our recent works [35,36], we proposed a synthetic method to prepare Ag/ZnO films via photo-fixation of Ag onto sol–gel-derived ZnO substrate, and the photocatalytic performance of the films was investigated in the removal of methylene blue and malachite green dyes under UV and visible light illumination. Here, we extend our efforts by exploring the possibility of constructing photocatalytic films from Ag-modified commercial ZnO powder as a simple and low-cost preparation method. Moreover, the degradation of PCA over pristine and Ag-modified ZnO films in distilled and drinking water under ultraviolet and direct sunlight is evaluated for the first time. It is found that under UV and natural sunlight in both types of water, the modified ZnO catalysts with various concentrations of silver ions demonstrate increased drug mineralization capabilities compared to pure ZnO. Finally, the possible photocatalytic mechanism of PCA degradation over Ag-modified ZnO films is discussed.

## 2. Materials and Methods

### 2.1. Materials

Zinc oxide commercial powder (≥99.0%), polyethyleneglycol 4000 (PEG 4000), ethanol ($C_2H_5OH$, ≥99.0%) and silver nitrate ($AgNO_3$) were received from Fluka (Buchs, Switzer-

land). Glass slide substrates (ca. 76 mm × 26 mm) were delivered from ISO-LAB (Schweitenkirchen, Germany).

For the photocatalytic tests, the commercially available medication paracetamol ($\lambda_{max}$ = 243 nm, 99.0%) from Teva was selected as a model contaminant.

Distilled and drinking waters were utilized in the photocatalytic experiments to demonstrate the degradation of paracetamol in the presence of various impurities as they occur in natural water systems. The drinking water in Sofia (Bulgaria) has a low content of dissolved salts, which makes it suitable for everyday use and useful for the normal functioning of cells in the body. The 'Beli Iskar' and 'Iskar' dams are where drinking water is mainly obtained from mountain water sources in Rila, which are over 2500 m above sea level. The mean value of the indicator "Total hardness" in the water supply system of the city of Sofia is <0.75 mgeq/L, which categorizes it as "freshwater". According to the requirements of the European legislation, the water from the "Beli Iskar" dam, regardless of the fact that it is mountainous and naturally clean, is also subject to purification and already passes through the purification station. For the destruction of microorganisms and dissolved organic substances, a disinfection process is carried out. Table 1 shows the major water quality indicators for Sofia.

**Table 1.** Characteristics of Sofia drinking water used in photocatalytic experiments.

| pH | $Na^+$, mg/L | $Ca^{2+}$, mg/L | $Mn^{2+}$, µg/L | $Fe^{2+}$, µg/L | $Cl^-$, mg/L | $SO_4{}^{2-}$, mg/L | $NO_3{}^-$, mg/L |
|---|---|---|---|---|---|---|---|
| 7.39 | <5.01 | <10.74 | <11 | <123 | <5 | <11 | <0.94 |

*2.2. Synthesis of ZnO and Ag/ZnO Powder Films*

The method which was employed for the powder catalysts that were manufactured is simple, affordable, well controlled and guarantees minimal energy use. First, 50 mL of ethanol was used as a solvent to dissolve 7 g of PEG 4000, a stabilizer. Magnetic stirring was used to stir the solution at 70 °C for 30 min, after which a clear-color solution was obtained. Commercial ZnO powder (7 g) was dissolved in 60 mL of ethanol over a 15 min period at room temperature. The resulting dispersion was combined with the PEG 4000 solution, which was then stirred for another 15 min and sonicated (15 kHz) for 30 min. The as-prepared white suspension was used to fabricate ZnO films.

The substrates for the pure and modified zinc oxide films were glass slides. They were cleaned with acetone and distilled water and dried for a while in the oven. After thoroughly reaching room temperature, the slides were rinsed with water and dried in the air. Using the dip-coating method, the glasses were submerged in the ZnO suspension at room temperature. Five coatings were used to fabricate four different types of films. After each layer, the samples were dried at 100 °C for 10 min. Finally, they were then annealed at 500 °C for 1 h in order to burn the organic material. After the production of ZnO films, the mass increase was 36.4 mg (8%). About 30 cm$^2$ of the surface of the glass substrates was covered.

Chemical photodeposition was used to create silver co-catalytically modified ZnO films. After being immersed in aqueous silver nitrate solution for 20 min, the Ag/ZnO films were photo-fixed (irradiated) with UV illumination and then washed with water. The modified films were dried at 100 °C for 10 min to remove nitrate ions. In our previous research, we successfully used the chemical photodeposition technique to photo-fix sol–gel films with silver ions and degrade organic dyes [35,36]. Experimental, structural and optical data demonstrated that the modified zinc oxide film (Ag $10^{-2}$/ZnO) has superior properties to pure zinc oxide. This allowed us to create and investigate the photocatalytic properties of two more additional films prepared with $10^{-3}$ M and $10^{-4}$ M AgNO$_3$ solutions (Ag $10^{-3}$/ZnO and Ag $10^{-4}$/ZnO).

### 2.3. Material Characterization

X-ray powder diffraction (XRD) analysis of pure and Ag/ZnO powder films was performed by an X-ray diffractometer (Siemens D500 with CuKα radiation, Karlsruhe, Germany) to examine the crystalline structure. The results were reported with counting time 2 s/step and over the angular range 20–80° at steps of 0.05° 2θ.

The shape and morphology of the powder samples were studied using scanning electron microscopy (SEM, Hitachi TM4000, accelerating voltage 15 kV, Krefeld, Germany). The elemental composition was stated by energy-dispersive X-ray spectroscopy (EDX) using Bruker AXS detector (Microanalysis GmbH, Berlin, Germany). X-ray photoelectron spectroscopy (XPS) analysis was used to determine the binding energy and the oxidation state of the nanostructures using ESCALAB MkII (VG Scientific, Manchester, UK) electron spectrometer with a base pressure in the analysis chamber of $5 \times 10^{-10}$ mbar, equipped with twin anode (MgKα/AlKα). The optic and photocatalytic properties of ZnO and Ag/ZnO films were characterized by ultraviolet–visible spectrophotometer (Evolution 300 Thermo Scientific (Madison, WI, USA).

Room-temperature photoluminescence (PL) of the samples was analyzed by using Varian Cary Eclipse spectrofluorimeter (Harbor, CA, USA) with an excitation wavelength of 325 nm.

### 2.4. Photocatalytic Test

The photocatalytic efficiency of nanostructures was estimated by measuring the degradation of paracetamol in distilled and drinking water. The standard main solution was made by dissolving 0.5 g of the drug in 0.5 l water. This solution was diluted to obtain working solutions with 50 ppm. Photocatalytic experiments were conducted with a photoreactor (200 mL volume) consisting of a magnetic stirrer (rotating speed controlled by stroboscope, 500 rpm) and an ultraviolet lamp (36 W, 315–400 nm emission range). All catalytic tests in the presence of UV illumination were conducted at room temperature (23 ± 2 °C).

For natural sunlight illumination, all experiments were performed in a similar photoreactor. The reactor was placed under direct sunlight irradiation for 4 h between 10:00 a.m. and 2:00 p.m. on sunny days of August 2023 in Sofia, Bulgaria. The reaction was performed at atmospheric conditions under constant stirring. The temperature was 34 ± 2 °C. Aliquot samples were taken at a certain time interval at all photocatalytic tests. In this way, the concentration of the drug was monitored using UV–vis spectrophotometer (Figure 1).

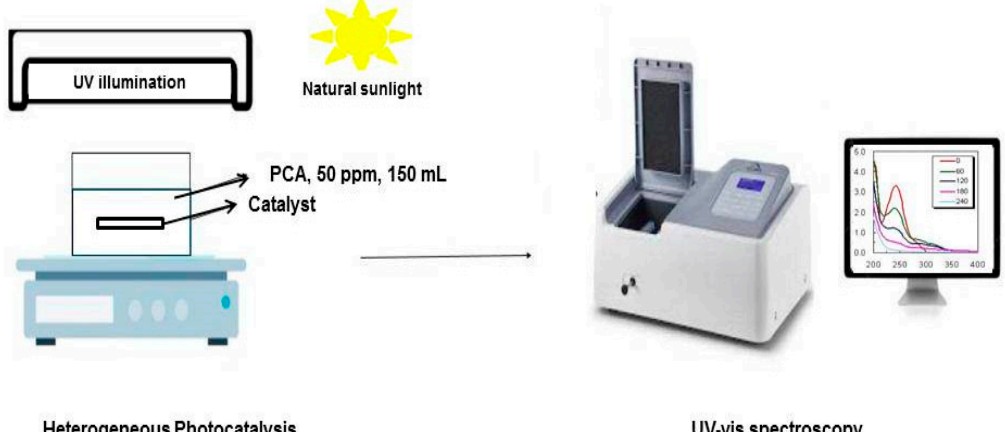

**Figure 1.** Heterogeneous photocatalysis involving a photoreactor and UV–vis absorption spectroscopy.

The percentage PCA degradation (D, %) was calculated using Equation (1).

$$\text{Paracetamol Degradation} = 100 \times \frac{C_0 - C}{C_0} \tag{1}$$

where $C_0$ is the initial drug concentration, C is the drug concentration in the reactor after illumination.

## 3. Results and Discussion

### 3.1. Physicochemical Characterization of the Nanostructure Materials

Different diffraction peaks at $2\theta = 31.94°$, $34.67°$, $36.51°$, $48.23°$, $56.84°$, $63.22°$, $67.53°$ and $68.18°$ are depicted in the diffraction pattern (Figure 2) of pure ZnO and Ag $10^{-2}$/ZnO powder films. These peaks correspond to the lattice plane orientations of each peak with hkl of (100), (002), (101), (102), (110), (103), (112) and (201), which are indexed to zinc oxide hexagonal wurtzite structure [37]. The peak positions are in good agreement with JCPDS card no. 96-230-0117. The powder films' diffraction peaks demonstrate a crystalline structure that is free of impurities or phase modifications. The strong and sharp peaks in the XRD patterns of both pure ZnO and Ag $10^{-2}$/ZnO samples indicate a high degree of crystallinity [38]. In addition, the diffraction pattern of Ag/ZnO (co-catalytically modified zinc oxide at the highest Ag⁺ concentration of $10^{-2}$M) shows additional small peaks. The crystal planes of metallic Ag with JCPDS card no. 96-901-3048 can be found at $2\theta = 38.46°$, $45.86°$ and $64.72°$, with diffraction peaks (111), (200) and (220) at $2\theta = 38.46°$, $45.86°$ and $64.72°$. The XRD patterns of pure ZnO and Ag $10^{-2}$/ZnO-modified films have no noticeable differences due to the low concentration of silver ions.

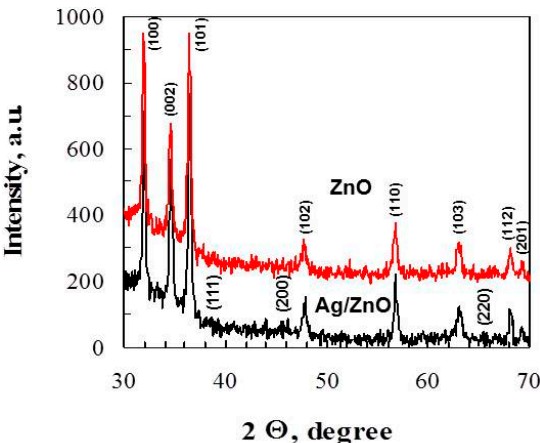

**Figure 2.** X-ray diffraction spectra of ZnO and Ag $10^{-2}$/ZnO films.

The Debye–Scherrer equation (Equation (2)) is employed to determine the crystallite size of both pure and modified films using the Bragg equation to determine the interpolation distance and crystallite length dimensions.

$$D_{hkl} = \frac{0.9\lambda}{\beta \cdot \cos\theta} \tag{2}$$

where $D_{hkl}$ is the average crystallite size (nm), $\lambda$ is the X-ray wavelength of CuK$\alpha$ radiation ($\lambda = 0.154056$ nm), $\beta$ is the full width at half maxima and $\theta$ is the Bragg's diffraction angle.

The introduction of Ag⁺ ions in ZnO has no significant effect on crystallite size. The average size of crystallites decreases slightly, with d(ZnO) being 52.6 nm and d(Ag $10^{-2}$/ZnO) being 47.2 nm. The parameters of the crystalline lattice of the pure and co-catalytically modified films are very close (ZnO: a = b = 3.2524 Å, c = 5.2124 Å, Ag $10^{-2}$/ZnO: a = b = 3.2518 Å, c = 5.2105 Å), confirming that the nanostructured materials possess a wurtzite hexagonal structure [39]. Furthermore, the c-axis lattice parameter is

used to determine the microstrain of the films. The calculation shows a positive value that indicates tensile strain. Compared to pure ZnO ($0.8 \times 10^{-3}$ a.u.), the microstrain in Ag $10^{-2}$/ZnO films ($0.7 \times 10^{-3}$ a.u.) has a slight magnitude of tensile strain.

Scanning electron microscopy measurements show that the surface of the ZnO and Ag $10^{-2}$/ZnO film presents a homogeneous morphology with particles of different sizes (Figure 3a,b). This is due to the inevitable agglomeration caused by the high temperature (500 °C) during preparation of the films. The fine granular structure of the film surface is revealed by higher-magnification images (see the inserts of Figure 3a,b). The silver co-catalytic modification does not significantly alter the morphology of the initial ZnO films (Figure 3b). As can be seen, the modification ions result in a small decrease in the crystal size, which is in line with the X-ray diffraction analysis. Figure 3c depicts a cross-section of the ZnO film, which also contains fine granules in the interior. The thickness of the films obtained after five coatings is 4 μm. After modification with silver ions, the powder film's thickness remains unchanged.

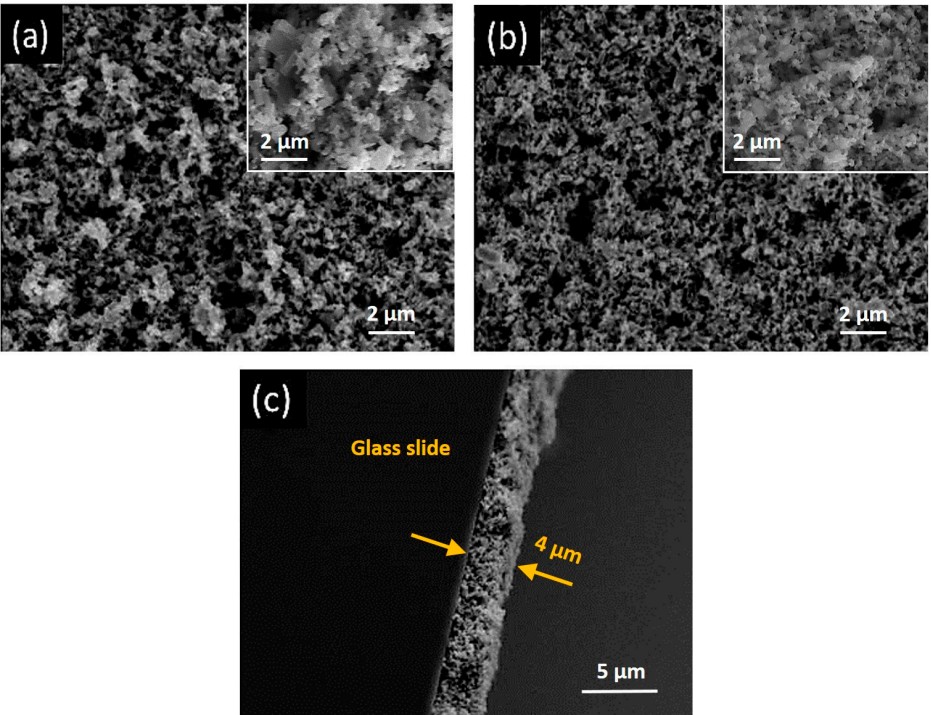

**Figure 3.** Surface morphology of (**a**) pure and (**b**) silver ($10^{-2}$ M) co-catalytically modified ZnO nanostructures. Cross-section of the ZnO film (**c**). The inserts of (**a**,**b**) show higher-magnification images.

Energy-dispersive X-ray spectroscopy is utilized to estimate the presence of Zn, O and Ag chemical elements (Figure 4) in the co-catalytically modified ZnO film with the greatest ($10^{-2}$ M) silver content.

The spectrum shows peaks with different strengths that correspond to the atoms of zinc, oxygen and silver. The weight percentage of Ag equals to 2.93 wt%. The absence of impurity peaks (other metal ions) in the EDS spectrum demonstrates the cleanliness of the powder films that are produced. This result demonstrates that high-purity Ag co-catalytically modified ZnO material photo-fixed under UV illumination could be successfully produced.

Figure 5a compares the full survey spectra of the ZnO and Ag $10^{-2}$/ZnO films, revealing the presence of Zn, O and adventitious carbon. The high-resolution Zn2p spectra of the samples are practically identical. Figure 5b shows two characteristic peaks positioned at 1021.2 eV and 1044.4 eV corresponding to the core levels of Zn $2p_{3/2}$ and Zn $2p_{1/2}$, respectively. These values are in agreement with those reported for pure ZnO [40]. Figure 5c

shows the details of the O 1s peaks deconvoluted using Gaussian lineshapes. Each peak is deconvoluted into three components at 530.7 eV ($O_I$), 532.6 eV ($O_{II}$) and 534.1 eV ($O_{III}$), corresponding to $O^{2-}$ ions in ZnO, oxygen vacancies and/or OH groups and physisorbed water molecules [41]. As seen, the components for both films show equal intensity proportions. This result suggests that the oxygen environment for ZnO and Ag $10^{-2}$/ZnO films is very similar. Interestingly, the HR scan in the 360–380 eV BE range for Ag $10^{-2}$/ZnO film (Figure 5d) detects two weak signals at 368.2 eV and 374.8 eV. These peaks can be attributed to the Ag $3d_{5/2}$ and Ag $3d_{3/2}$, supporting the presence of metallic Ag [42]. Contrary to this, XPS examination of ZnO film does not show the existence of Ag.

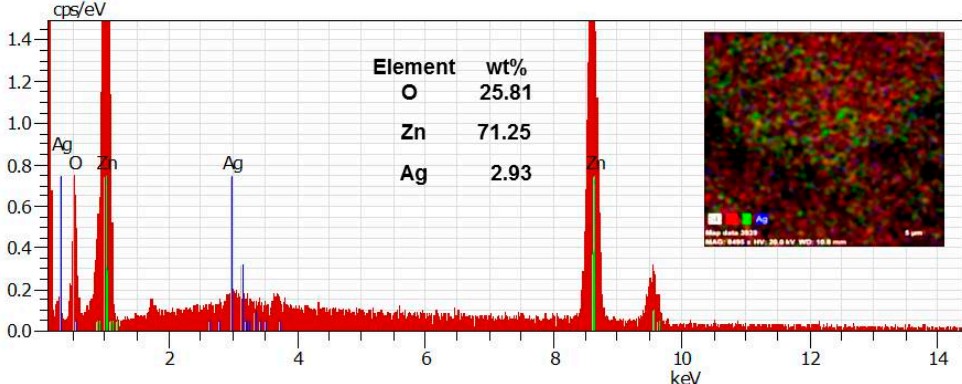

**Figure 4.** Energy-dispersive X-ray spectroscopy of silver ($10^{-2}$ M) co-catalytically modified zinc oxide powder film.

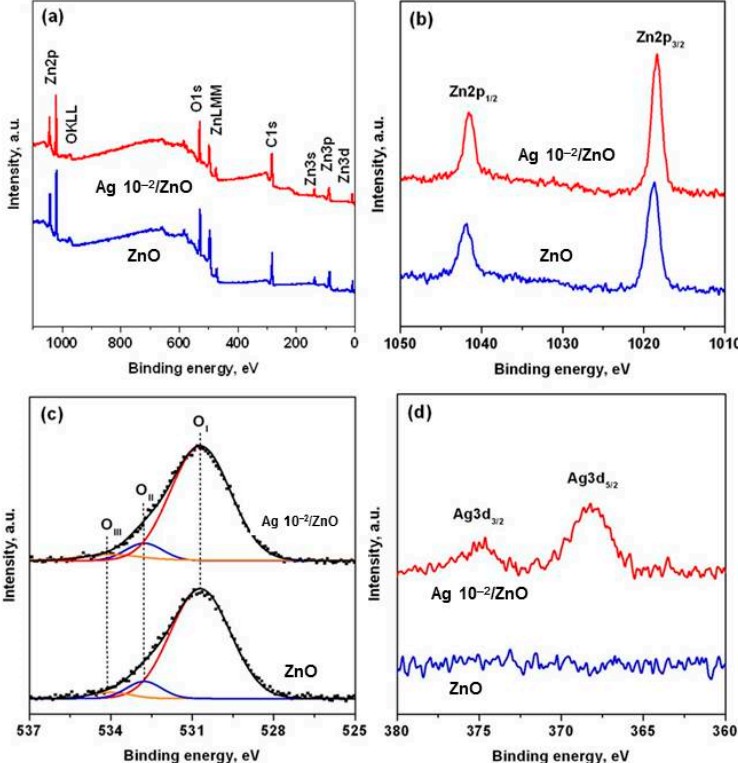

**Figure 5.** (**a**) XPS survey spectra, (**b**) Zn2p, (**c**) O 1s and (**d**) Ag 3d core-level spectra of the ZnO and Ag $10^{-2}$/ZnO films.

UV–vis absorbance spectroscopy is used to calculate the band gap energy ($E_g$) of pure and co-catalytically modified ZnO films (Figure 6a). The spectra show the presence of a

band edge in the ultraviolet region ($\lambda_{maxAg\,10^{-2}/ZnO}$ = 365 nm and $\lambda_{maxZnO}$ = 362), which is consistent with zinc oxide's properties [43]. This is explained by the zinc oxide being photoexcited from the lower valence band to the higher conduction band. Additionally, the reaction between the noble metal and semiconductor is assumed to be responsible for the shifting of absorbance peaks to higher wavelengths [44,45]. Using data from UV–vis spectroscopy, the band gap energy ($E_g$) is computed (Figure 6b).

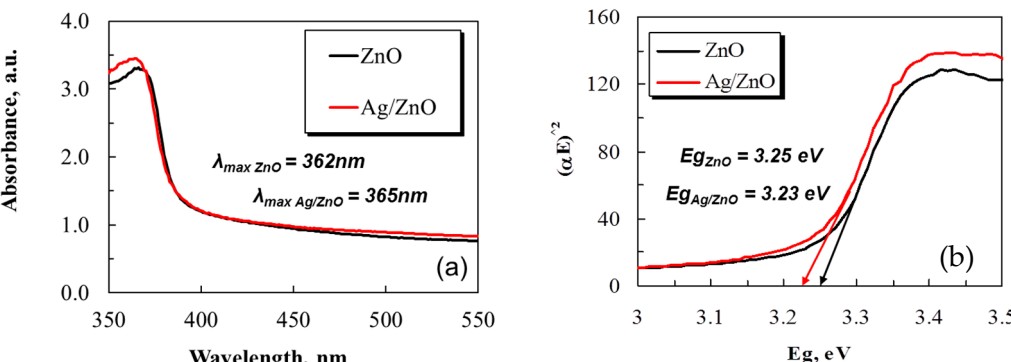

**Figure 6.** (**a**) UV–vis absorption spectra and (**b**) Tauc plots of ZnO and Ag $10^{-2}$/ZnO powder films.

The Tauc equation (Equation (3)) yields the band gap energy values.

$$(\alpha h\nu)^2 = A\left(h\nu - E_g\right) \tag{3}$$

where $\alpha$ is the absorption coefficient, h is the Planck's constant, $\nu$ is the photon frequency, A is the constant and $E_g$ is the energy band gap. The Eg values are calculated by extrapolating a straight line to the x-axis from plots of $(\alpha h\nu)^2$ versus hv. Figure 6 illustrates the values obtained from the Tauc extrapolation vs. the energy band gap of the films and demonstrates the alteration in the absorption edge that is attributed to the presence of $Ag^+$ in the ZnO [46]. As demonstrated in Figure 6b, pure ZnO has a band gap energy of 3.25 eV, while silver co-catalytically modified films have a lower $E_g$ of 3.23 eV (Ag $10^{-2}$/ZnO). Since they can trap more electrons, the silver co-catalytically modified ZnO films made with the aid of UV light (Ag $10^{-2}$/ZnO) are anticipated to be more effective in the oxidation and reduction reactions that take place on the catalyst.

Figure 7 compares the PL spectra of ZnO and Ag $10^{-2}$/ZnO films. The lineshapes of the spectra are very similar. Both samples possess UV PL below 400 nm, ascribed to the intrinsic near-band edge (NBE) emission of ZnO [47]. Visible emission bands in ZnO are usually assigned to the defect-related recombinations, such as zinc vacancy ($V_{Zn}$), zinc interstitial ($Zn_i$), oxygen vacancy ($V_O$) or interstitial oxygen ($O_i$) [48]. The violet emission located at 423 nm for ZnO and Ag $10^{-2}$/ZnO is related to the recombination of an electron in the defect state of Zni with a hole in the valence band [48]. The presence of a blue band at 444 nm is caused by transitions between the extended Zni levels (singly or doubly ionized $Zn_i$) and the valence band [49]. Notably, the intensity of ZnO PL features decreases after introducing Ag. This result indicates the inhibition of photogenerated electron–hole pair recombination in the Ag $10^{-2}$/ZnO film.

### 3.2. Effect of $Ag^+$ Concentration and Type of Water

Initially, we investigate the photocatalytic properties only of zinc oxide modified with silver ions at a concentration of $10^{-2}$ M. The experimental results show that the silver-modified sample has higher activity compared to the pure semiconductor. Furthermore, the photocatalytic characteristics of two additional films (Ag $10^{-3}$/ZnO and Ag $10^{-4}$/ZnO) are also studied. Adsorption tests conducted in complete darkness show that 5% of the analgesic is adsorbed before being exposed to light radiation. The pseudo-first-order kinetics of contaminant removal is confirmed by the logarithmic plots of paracetamol concentration versus irradiation time, and the rate constants (k) are shown in Figure 8a,b. It

is found that the mean time, during the co-catalyst fixation process, and the photocatalytic properties for Ag-modified films rise proportionally with the $Ag^+$ ions concentration. This is supported by the fact that the rate constant, k, increases as the concentration of silver ions increases from $10^{-4}$ to $10^{-2}$ M in both cases (paracetamol in drinking water and distilled water). Since UV light can excite electrons in the catalyst and oxidize the organic pollutant, higher values of the rate constants indicate faster PCA degradation. This finding may also be explained by a decrease in the rate of recombination of the formed charge pairs ($e^-/h^+$) and a narrowing of the band gap width in the samples with Ag.

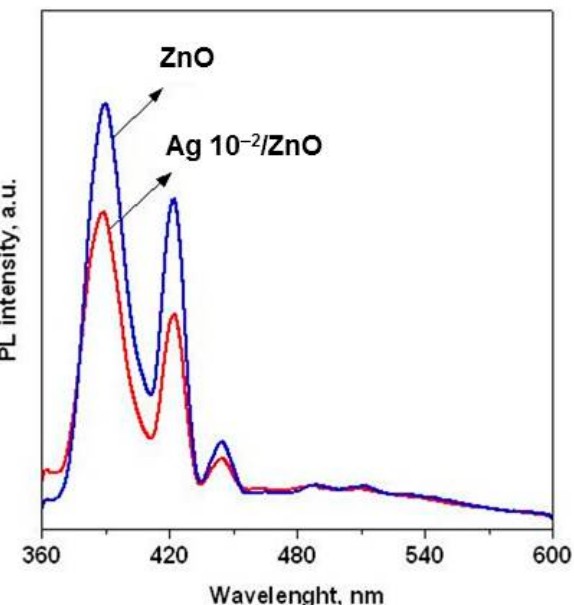

**Figure 7.** Room temperature PL spectra of ZnO and Ag $10^{-2}$/ZnO powder films.

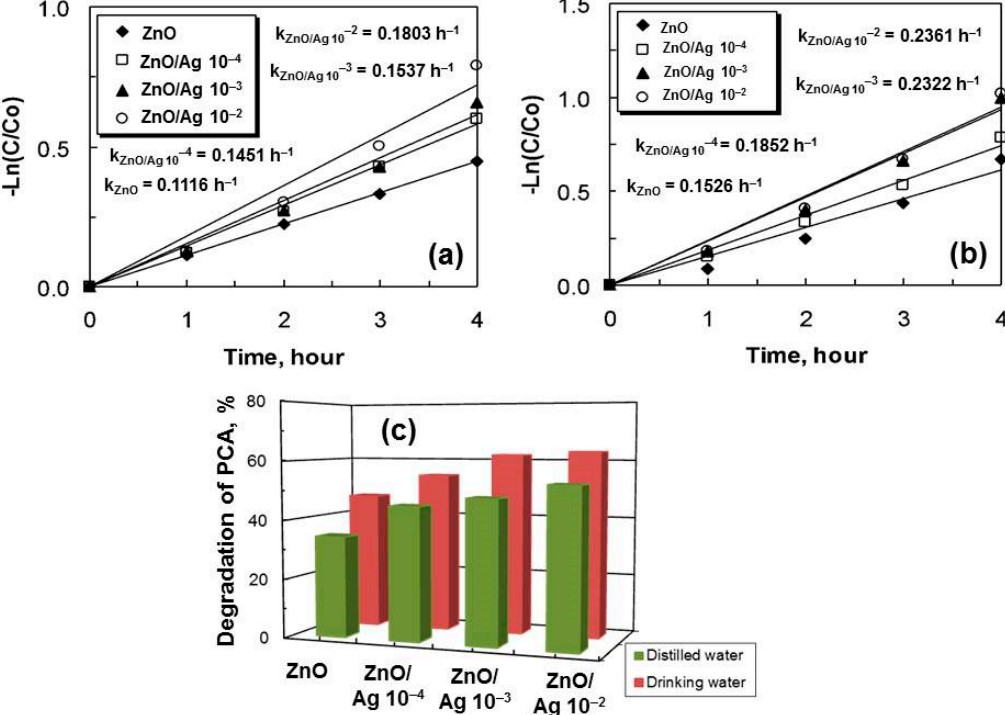

**Figure 8.** Kinetics of removal of paracetamol in distilled (**a**) and drinking (**b**) water; the percent degradation (**c**) of the drug under UV illumination.

When UV or sunlight is absorbed by ZnO, electrons in the valence band (VB) are excited to move into the conduction band (CB). The rate of photocatalysis is decreased by the recombination of $e^-/h^+$ charge carriers. The ability of Ag to trap electrons may help to explain the beneficial effect of co-catalytic modification with silver ions on ZnO efficiency for paracetamol photodegradation. The freshly formed Fermi level of Ag/ZnO is lower than the energy of the bottom of the CB (Fermi level of Ag 0.99 eV vs. NHE) [50], and the transfer of photogenerated $e^-$ from the CB into Ag is possible. The drug could be broken down by the highly reactive species (superoxide anion) produced when oxygen molecules interact with the electrons in Ag. According to Ramasamy et al. [51], the electron transfer to the silver ion cannot be effectively competed with by oxygen-scavenging electrons at the surface of excited semiconductor particles. As a result, the formation of $O_2^{\bullet-}$ is decreased because the electron transfer to the silver ion is quicker than the electron transfer to the oxygen molecule. However, the loading of Ag metals on the ZnO surface in the silver co-catalytically modified semiconductor can hasten the transport of photogenerated electrons to the outer systems. Deposits of metal become partially negatively charged as a result of electron transfer. By accelerating the transfer of electrons to dissolved oxygen molecules, the deposits of silver ions on the surface increase photoactivity. As a result, oxygen reduction leads to the formation of the superoxide anion radical through the transfer of trapped electrons from Ag metal to oxygen. Furthermore, the holes in VB of ZnO interact with water molecules to create hydroxyl radicals, which also break down the organic molecules. A schematic of the proposed photocatalytic mechanism of Ag–ZnO film is presented in Scheme 1.

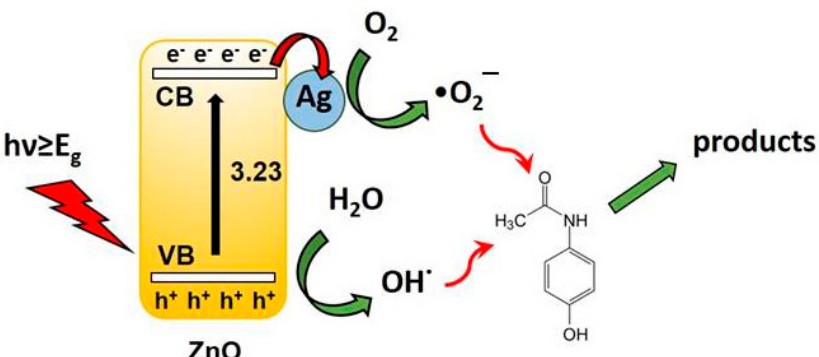

**Scheme 1.** Photocatalytic mechanism of Ag–ZnO film for photocatalytic degradation of paracetamol under light illumination.

The following Equations from (4) to (8) describe the electron–hole reaction in the Ag/ZnO nanostructure, which breaks down organic pollutants such as drugs.

$$Ag/ZnO + h\nu \rightarrow e^- + h^+ \tag{4}$$

$$e^- + O_2 \rightarrow O_2^{\cdot-} \tag{5}$$

$$h^+ + H_2O \rightarrow H^+ + OH^{\cdot} \tag{6}$$

$$h^+ + OH^- \rightarrow OH^{\cdot} \tag{7}$$

$$PCA + O_2^{\cdot-}/OH^{\cdot} \rightarrow \text{degradation products} \tag{8}$$

By restricting $e^-/h^+$ recombination by charge separation and favoring interfacial charge transfer between ZnO and Ag, the co-catalytic modification of zinc oxide with silver ions enhances the semiconductor's photocatalytic properties.

Figure 8c illustrates the percentage of pharmaceutical removal using four different catalysts, supporting the accuracy of the rate constants. The Ag $10^{-2}$/ZnO film displays the highest activity after being exposed to UV light for 4 h in both drinking and distilled water. The pure and Ag-modified nanostructure films exhibit a lower degradation rate of paracetamol in distilled water in comparison with in drinking water. This observation

can be attributed to the variations of pH values. It is well known that pH regulates the photocatalytic process. The pH levels of drinking water and distilled water are different from one another, albeit only by a little ($pH_{distilled}$ = 6.8 and $pH_{drinking}$ = 7.39). Ramasamy et al. [51] studied the impact of pH on the breakdown of paracetamol in greater detail. They discovered that as the pH rose from 4 to 8.5, the rate of drug degradation accelerated. Electrostatic forces between the contaminants and the catalyst can account for this. The catalyst surface is negatively charged at pH levels higher than the point of zero charge and vice versa. According to published research, the pH point of Ag/ZnO at zero charge has the same value as ZnO, which is 9.0 [52]. Paracetamol has an acid ionization constant of 9.5. Therefore, a rise in pH above 9 gradually accelerates the electrostatic attraction between the Ag/ZnO surface and contaminants, which has a negative impact on mineralization and lowers the rate constant. At pH = 7.39 (drinking water), paracetamol's maximum rate constant of 0.2361 $h^{-1}$ is seen, and, as pH falls even by a little, the rate constant (k) decreases to 0.1803 $h^{-1}$. Our results are in accord with previous studies [53].

### 3.3. Effect of Type Light Illumination—Ultraviolet and Natural Sunlight

In the presence of a suitable photocatalyst, sunlight, an abundant natural source of energy, can be used in the photocatalytic treatment of drugs, improving the process' economic viability [54]. Under ultraviolet light, the synthesized Ag/ZnO nanostructure films are discovered to be photocatalytically active, and sunlight is used to test their applicability in photocatalysis. Figure 9 shows that drug removal can also occur when sunlight is present. The graph of $-Ln(C/Co)$ vs. time also shows that Ag/ZnO prepared at $10^{-2}$ M silver ion concentration has the highest photocatalytic efficiency in both types of waters, while the other samples exhibit a decline in efficiency. The rate constant comparison between pure and modified films containing Ag($10^{-4}$ M), Ag($10^{-3}$ M) and Ag($10^{-2}$ M) is provided in Figure 9a,b. The same tendency is seen here as well, namely, that paracetamol in drinking water breaks down more quickly than in distilled water. The impact of the type of irradiation is another effect that we are able to estimate in this work—62% of the paracetamol in drinking water can be degraded using UV illumination, and around 83% is degraded in 240 min under natural sunlight irradiation in the presence of the Ag $10^{-2}$/ZnO powder films (Figures 8c and 9c). This demonstrates that the Ag/ZnO films exhibit good photocatalytic activity under natural sunlight and ultraviolet light. The addition of silver ions to ZnO improves the catalyst's ability to block the recombination of electron–hole pairs and expands the range of solar spectrum wavelengths that can be absorbed by the catalyst, improving the activity of the nanostructures in drug degradation by sunlight. Under sunlight, the nanocomposites that exhibit light absorption across the entire spectrum exhibit strong photocatalytic activity. We assume that the co-catalyst experiences additional excitation from natural sunlight causing activation of the ZnO surface active centers. This hypothetical reason can be used to explain why pure and silver co-catalytically modified zinc oxide exhibits higher activity in the presence of natural sunlight. For more information and a clearer understanding of this claim, more research will be required in the future.

### 3.4. Effect of Recycle Times of Pure and Ag Co-Catalytically Modified ZnO Powder Films on the Photocatalytic Degradation of Paracetamol

A green technology known as heterogeneous photocatalysis typically has no issues with waste disposal. Therefore, it is essential to maintain high catalytic activity throughout each usage cycle. The regeneration and reuse of ZnO and Ag $10^{-2}$/ZnO films are investigated, and the outcome is presented in Figure 10. It is clear that after each cycle, the nanostructures' photocatalytic activity slightly declines. The photocatalytic degradation of the drug is reduced by about 2–3% after three cycles for both types of catalysts. The results confirm that Ag ($10^{-2}$ M) co-catalytically modified film remains highly photocatalytically active after three cyclical experiments, and, furthermore, the SEM image (Figure 10e) of the film after these measurements reveals that the integrity and surface morphology of the starting material are retained.

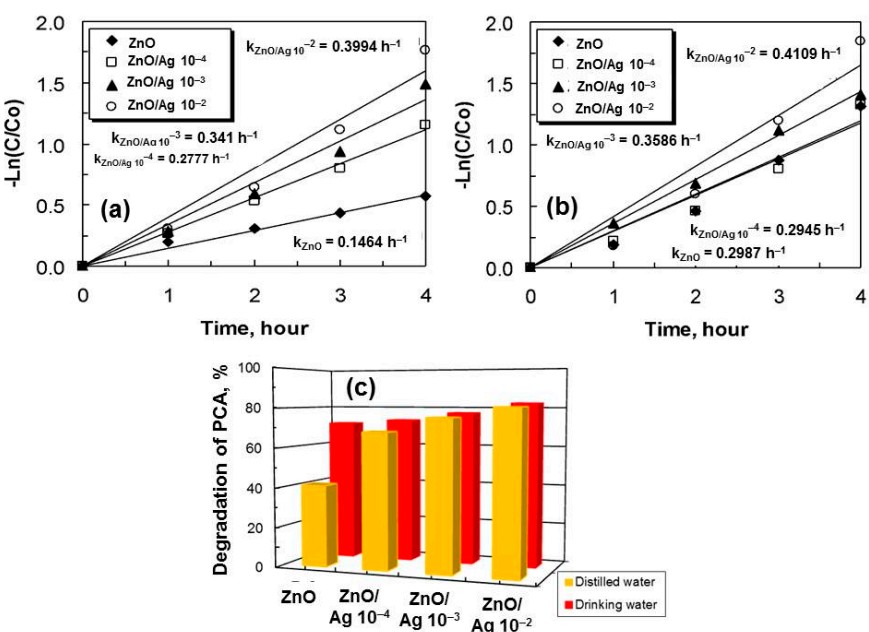

**Figure 9.** Kinetics of removal of paracetamol in distilled (**a**) and drinking (**b**) water; the percent degradation (**c**) of the drug under natural sunlight illumination.

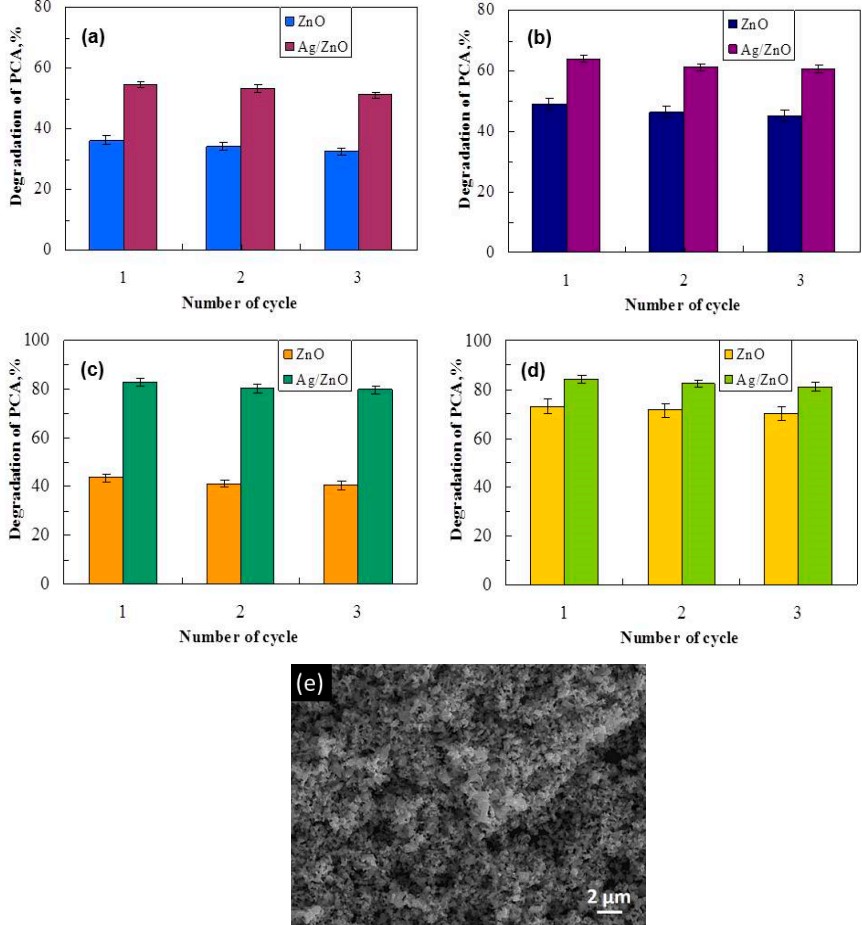

**Figure 10.** Photostability of pure and Ag $10^{-2}$ M/ZnO photocatalysts on the removal of paracetamol using UV ((**a**) distilled and (**b**) drinking water) and natural sunlight ((**c**) distilled and (**d**) drinking water). (**e**) SEM image of the Ag $10^{-2}$ M/ZnO film after 3 cyclical experiments.

Finally, Figure 11 displays UV–vis spectra of paracetamol degradation in the presence of an Ag $10^{-2}$/ZnO photocatalyst, as obtained under natural sunlight and UV light in drinking and distilled water. As previously reported [55], the spectrum of paracetamol shows bands at 194 and 243 nm due to the $\pi \rightarrow \pi^*$ and to the $n \rightarrow \pi^*$ electronic transitions of the aromatic ring and the C=O group, respectively [55]. As can be seen, these bands gradually decline with irradiation time, indicating the efficient photocatalytic reactions. According to the literature, photocatalytic degradation of paracetamol leads to the formation of non-toxic carboxylic acids as final products of the process. Since no additional bands appear in the spectra after the photocatalytic reactions, one may conclude that the presence of toxic by-products in the final solutions can be ruled out.

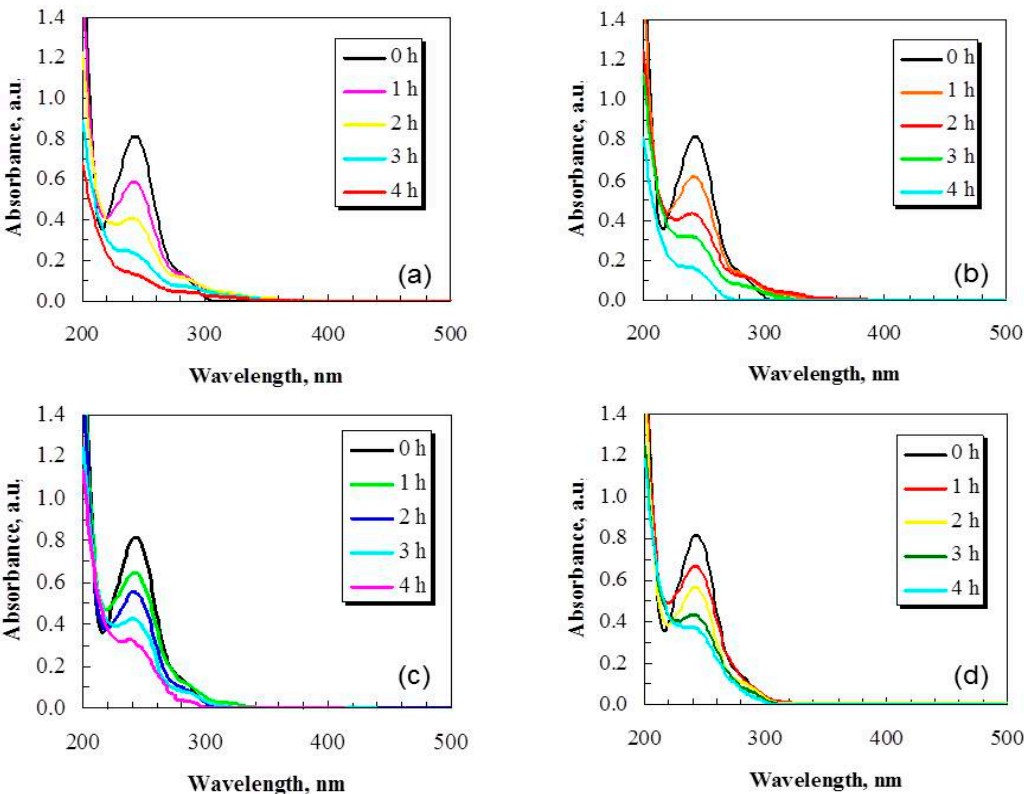

**Figure 11.** Absorbance spectra of the degradation of paracetamol under natural sunlight (drug in drinking (**a**) and distilled (**b**) water) and ultraviolet (drug in drinking (**c**) and distilled (**d**) water) illumination using Ag $10^{-2}$/ZnO powder film.

## 4. Conclusions

ZnO powder films were produced through a process that is economical, environmentally friendly and green. Namely, ZnO nanostructures were modified with different concentrations of silver ions ($10^{-2}$, $10^{-3}$ and $10^{-4}$ M) via photo-fixation with ultraviolet (UV) illumination. It was found that the surface morphology of the initial and Ag-modified films was homogeneous and uniform. The pseudo-first-order kinetic model accurately depicted the photocatalytic degradation of paracetamol. According to the photocatalytic results, adding silver ions to ZnO improves the ability of paracetamol to break down in distilled and drinking water under ultraviolet and direct sunlight irradiation. The improvement of the oxidizing power of ZnO is due to electron transfer from its conduction band to Ag and the inhibition of photogenerated electron–hole pair recombination. Under sunlight exposure, the Ag $10^{-2}$/ZnO films achieved the highest degradation percentages for paracetamol in both distilled and drinking waters ($D_{distilled}$ = 82.86%, $D_{drinking}$ = 84.1%). Under UV exposure, the degradation percentages were slightly lower but still higher than those achieved by pure zinc oxide ($D_{distilled}$ = 54.7%, $D_{drinking}$ = 63.87%). The enhanced

photocatalytic efficiency, excellent photostability and positive impact of silver ions make this system an attractive option for the degradation of pharmaceutical drugs, contributing to the advancement of eco-friendly and efficient wastewater treatment technologies.

**Author Contributions:** D.I.: Photocatalytic investigations and data curation; G.T.: XPS analysis and data curation; N.K.: Conceptualization, methodology, data curation, writing—original draft. All authors have read and agreed to the published version of the manuscript.

**Funding:** This research was funded by the Bulgarian NSF project KP-06-N59/11 (КП-06-Н59/11) and by the European Union—NextGenerationEU through the National Recovery and Resilience Plan of the Republic of Bulgaria, project no. BG-RRP-2.004-0008.

**Data Availability Statement:** Not applicable.

**Acknowledgments:** The authors are grateful to financially supported by the Bulgarian NSF project KP-06-N59/11 (КП-06-Н59/11) and the European Union—NextGenerationEU through the National Recovery and Resilience Plan of the Republic of Bulgaria, project no. BG-RRP-2.004-0008.

**Conflicts of Interest:** The authors declare no conflict of interest.

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
