# Peer review of "Degradation of Paracetamol in Distilled and Drinking Water via Ag/ZnO Photocatalysis under UV and Natural Sunlight"

_water, doi:10.3390/w15203549_

Round 1

Reviewer 1 Report

The paper can be reconsidered for publication after modified the items below, please note that the all items should be modified precisely.

1. The Introduction section is too simple, more background and motivation should be added.

2. The abstract should contain some quantitative information also.

3. The novelty of the work should be described in the introduction.

4. In order to increase the quality of literature, the recently published relate to this paper should be cited in the reference part:

§  Surfaces and Interfaces Volume 40, August 2023, 102970

 §  Inorganic Chemistry Communications Volume 153, July 2023, 110823

§  Surfaces and Interfaces Volume 38, June 2023, 102830

§  Journal of the Taiwan Institute of Chemical Engineers Volume 149, August 2023, 105004

--

Author Response

Response to Reviewer 1

Ms. Ref. No.: 2646932

Title: “Degradation of Paracetamol in Distilled and Drinking Water via Ag/ZnO Photocatalysis Under UV and Natural Sunlight”, written by Dobrina Ivanova, George Tzvetkov, Nina Kaneva

            Water

We thank very much to the Reviewer for the valuable marks, comments and suggestions, supporting the presentation of our work. We accept all the recommendations and corrected our paper properly following the suggestions given by the Reviewer in the same sequence.

Hope you find revised manuscript suitable for publication in Water. We look forward to hearing from you in due course.

Reviewer 1: The paper can be reconsidered for publication after modified the items below, please note that the all items should be modified precisely.

  1. The Introduction section is too simple, more background and motivation should be added.

We changed the Introduction, as you recommended.

“Various pharmaceutical pollutants have been detected in wastewater [1, 2], surface and groundwater [3, 4], and even in drinking water [5, 6]. More specifically, antibiotics, analgesics, antipyretics and hormones are frequently found in the aquatic environment [2, 7]. As long-term consumption of pharmaceutical contaminants through drinking water results in carcinogenetic disorders that have a negative impact on the human health, the presence of these compounds in drinking water is a significant problem and causes public concern [6]. The harmful effects of pharmaceuticals include toxicity, resistance to pathogenic microorganisms, genotoxicity, and endocrine disruption [8, 9].

One of the most widely used medications – paracetamol (acetaminophen) is a common, non-biodegradable and highly water-soluble compound that is present in more than hundred pharmaceutical products [10]. Paracetamol (PCA) is frequently used to treat headaches, a variety of colds and influenza. According to the report from 2000, pa-racetamol ranks third among the top medications and its consumption exceeds 400 tons annually [11]. This analgesic has been discovered in natural and European wastewaters in amounts up to 10 g/L and 6 g/L, respectively [12]. Recently, many methods, including electrochemical [13, 14], ozonation [15, 16], Fenton and photo-Fenton [17, 18], and H2O2/UV oxidation techniques [19, 20] to mineralize PCA have been reported. However, one of the most promising AOPs (advanced oxidation processes) for the removal of PCA from water medium is heterogeneous photocatalysis.

Heterogeneous photocatalysis with semiconductor materials like zinc oxide (ZnO) has indeed gained popularity as an effective method for treating wastewater contami-nated with dyes, pesticides, and pharmaceuticals [10]. ZnO is a low-cost, non-toxic, chemically stable and easy to produce material with a range of applications, including micro-gas sensing devices [19], solar cells [20], nonlinear optics [21], integrated photonic devices [22] and photocatalysts for wastewater treatment [23, 24]. However, under visible or sunlight irradiation ZnO cannot be used as a catalyst due to its significant band gap energy. Additionally, due to the rapid recombination rate of charge (e-/h+) couples, its photocatalytic efficiency is also constrained. To overcome this problem, ZnO surface can be modified and functionalized with co-catalysts, typically noble metals as Pt, Pd, Ag, and Au, and based on the formation of heterojunctions to limit the possibilities for re-combination [25–27]. Different approaches have been used to combine the ZnO substrate with metallic co-catalysts [25-27]. Among them, the direct photofixation or photocatali-tycally deposition of metal clusters under UV light gained attention recently [28, 29]. More precisely, under the influence of the photogenerated charges on the ZnO substrate, the co-catalyst can be deposited as a result of the reduction of metal ions, which are typically from an aqueous solution [28, 29]. As photogenerated electron acceptors, the noble metal ions (Ag+, Pt2+, Pd2+, Au3+) can be successfully reduced to the corresponding metals [30-32].

Most of the studies that have examined ZnO and co-catalytically modified ZnO focus on the systems using suspended photocatalyst particles. This limits their practical applicability due to the requirement of centrifugation or filtration to reuse fine ZnO par-ticles. The design of systems where the photocatalysts are supported on inert substrates as layers and thin films proved to be successful in overcoming such obstacles [30]. Very recently, Hao and co-workers [33] described the construction of hydrothermally grown ZnO films on wire mesh, subsequently decorated with Ag nanospheres using the im-pregnating photoreduction treatment. The formed Ag–ZnO heterojunction improves the absorption of UV and visible light, thereby boosting the photocatalytic properties of the films. Rati and co-workers [34] reported that Ag addition contributes to the improvement of ZnO thin films throughout the photocatalytic process of Methylene blue degradation. Therefore, developing novel ZnO-based thin film photocatalysts for water remediation is of constant interest.

The main goal of the present work is to elaborate a new fabrication strategy to grow Ag-modified ZnO films for photocatalytic removal of paracetamol. In our recent work [35, 36], we proposed a synthetic method to prepare Ag/ZnO films via photo-fixation of Ag onto sol-gel derived ZnO substrate and the photocatalytic performance of the films have been investigated in the removal of Methylene blue and Malachite green dyes under UV and visible light illumination. Here, we extend our efforts by exploring the possibility to construct photocatalytic films from Ag-modified commercial ZnO powder as a simple and low-cost preparation method. Moreover, the degradation of PCA over pristine and Ag-modified ZnO films in distilled and drinking water under ultraviolet and direct sun-light was evaluated for the first time. It was found that under UV and natural sunlight in both types of waters, the modified ZnO catalysts with various concentrations of silver ions demonstrated increased drug mineralization capabilities compared to pure ZnO. Finally, the possible photocatalytic mechanism of PCA degradation over Ag-modified ZnO films is discussed.”

  1. The abstract should contain some quantitative information also.

We added quantitative description in the abstract: “…Our experimental evidences showed that the Ag/ZnO nanostructure films are much more active than pristine ZnO films in the photodegradation process. Namely, the photocatalytic efficiency of the films modified with 10-2 M concentration of silver ions achieved the highest degradation (D) percentages for paracetamol at both types of waters (Ddistilled = 80.97%, Ddrinking = 82.5%) at natural sunlight. Under UV exposure, the degradation percentages are slightly lower but still higher than those achieved by pure ZnO films (Ddistilled = 53.13%, Ddrinking = 61.87%). It has been found that the photocatalytic activity grows in direct proportion to the concentration of Ag+ ions: ZnO < Ag,10-4/ZnO, < Ag,10-3/ZnO < Ag,10-2/ZnO…..”

  1. The novelty of the work should be described in the introduction.

In the present paper, we modify with silver ions films prepared from commercial ZnO powder for the first time. We believe that this fabrication strategy is much faster and cost-effective. The use of sunlight to break down paracetamol in drinking and distilled water is also a novelty in this study. All these effects have not previously been taken into account.

The explanation is given in the introduction: “The main goal of the present work is to develop a new fabrication strategy to grow Ag-modified ZnO films for photocatalytic paracetamol removal of paracetamol. In our recent work [35, 36], we proposed a successful strategy to develop Ag/ZnO films via photo-fixation of Ag onto sol-gel derived ZnO substrate and the photocatalytic performance of the films have been investigated in the removal of Methylene blue dye under UV and visible light illumination. Here, we extend our efforts by exploring the possibility to construct photocatalytic films from Ag-modified commercial ZnO powder as a simple and low-cost preparation method. Moreover, the degradation of PCA over pristine and Ag-modified ZnO films in distilled and drinking water under ultraviolet and direct sun-light was evaluated for the first time. It was found that under UV and natural sunlight in both types of waters, the modified ZnO catalysts with various concentrations of silver ions demonstrated increased drug mineralization capabilities compared to pure ZnO. Finally, the possible photocatalytic mechanism of PCA degradation over Ag-modified ZnO films was discussed.”

  1. In order to increase the quality of literature, the recently published relate to this paper should be cited in the reference part:
  • Surfaces and Interfaces Volume 40, August 2023, 102970
  • Inorganic Chemistry Communications Volume 153, July 2023, 110823
  • Surfaces and Interfaces Volume 38, June 2023, 102830
  • Journal of the Taiwan Institute of Chemical Engineers Volume 149, August 2023, 105004

We added these reports in the introduction.

Mahdi, A.; Obeid, R.; Abdullah, K.; Mohammed, S.; Kadhim, A.; Ramadan, M.; Hussien, B.; Alkahtani, A.; Ali, F.; Alkhathami, A.; Al-Fatolahi, L.; Fakhri, A. A facile construction of NiV2O6/CeO2 nano-heterojunction for photo-operated process in water remediation reaction, antibacterial studies, and detection of D-Amino acid in peroxidase system. Surf. Sci. 2023, 40, 102970.

Aldhalmi, A.; Alkhayyat, S.; Albahadly, W.; Jawad, M.; Alsaraf, K.; Muedii, Z.; Ali, F., Ahmed, M.; Asiri, M.; Al-Fatolahi, L.; Fakhri, A. A novel fabricate of iron and nickel-introduced bimetallic MOFs for quickly catalytic degradation via the peroxymonosulfate, antibacterial efficiency, and cytotoxicity assay. Inorg. Chem. Commun. 2023, 153, 110823.

Chen, Y.; Jihad, A.; Hussam, F.; Al-Abdeen, S.; Hussein, J. ; Adhab, Z.; Alzahraa, Z.; Ahmad, I.; Fatolahi, L.; Janani, B. A facile preparation method for efficiency a novel LaNiO3/SrCeO3 (p-n type) heterojunction catalyst in photocatalytic activities, bactericidal assessment and dopamine detection. Surf. Int. 2023, 38, 102830.

Syed, A.; Elgorban, A.; Bahkali, A.; Eswaramoorthy, R.; Verma, M.; Varma, R.; Janani, B. Highly-impressive performances of novel NiCo2O4/Bi2O3/Ag2ZrO3 nanocomposites in photocatalysis system, removal pathway, toxicity estimation, and antibacterial activities. J. Taiwan Inst. Chemical. Eng. 2023, 149, 105004.

In the References, these sources are marked in blue.

Reviewer 2 Report

The manuscript is within the scope of the journal and the topics is of interest for the readers. The research need is clear but the state of the art on the development of Ag/ZnO nanostructure films  is not well described. this materials have been already used and the novelty of the work is not well described. It seems to be the Ag load but this point is critical to assess the novelty of the manuscript. 

The methodology seems to be similar to the one used in other publications. If that is the case the authors should be referenced and if not the main differences need to be identified and justified. If significant different materials with respect to the state of the art are obtained, it needs to be highlighted.

How can be the authors be sure that intermediate oxidation subproducts are not formed?

What is the final toxicity of the treated samples?

Adequate

Author Response

Response to Reviewer 2

Ms. Ref. No.: 2646932

Title: “Degradation of Paracetamol in Distilled and Drinking Water via Ag/ZnO Photocatalysis Under UV and Natural Sunlight”, written by Dobrina Ivanova, George Tzvetkov, Nina Kaneva

            Water

We thank very much to the Reviewer for the valuable marks, comments and suggestions, supporting the presentation of our work. We accept all the recommendations and corrected our paper properly following the suggestions given by the Reviewer in the same sequence.

Hope you find revised manuscript suitable for publication in Water. We look forward to hearing from you in due course.

Reviewer 2: The manuscript is within the scope of the journal and the topics is of interest for the readers.

  1. The research need is clear but the state of the art on the development of Ag/ZnO nanostructure films is not well described. This materials have been already used and the novelty of the work is not well described. It seems to be the Ag load but this point is critical to assess the novelty of the manuscript.

The methodology seems to be similar to the one used in other publications. If that is the case the authors should be referenced and if not the main differences need to be identified and justified. If significant different materials with respect to the state of the art are obtained, it needs to be highlighted.

In our previous two studies - Ivanova, D.; Mladenova, R.; Kolev, H.; Kaneva, N. Effect of Ultraviolet Illumination on the Fixation of Silver Ions on Zinc Oxide Films and their Photocatalytic Efficiency. Catalysts 2023, 13, 1121.; Kaneva, N.; Bojinova, A.; Papazova, K. Enhanced Removal of Organic Dyes Using Co-Catalytic Ag-Modified ZnO and TiO2 Sol-Gel Photocatalysts, Catalysts 2023, 13, 245., we synthesize sol-gel thin films (ZnO, TiO2) using the following reagents – 2-methoxyethanol, monoethanolamine, zinc acetate dehydrate, titanium isopropoxide. The method of photo-fixation and the method of modification with silver ions is the same. In these two papers, we degrade the organic dyes Malachite Green and Methylene Blue in the presence of ultraviolet, visible light and in the dark.

In the present paper, we modify with silver ions films prepared from commercial ZnO powder for the first time. We believe that this fabrication strategy is much faster and cost-effective. The use of sunlight to break down paracetamol in drinking and distilled water is also a novelty in this study. All these effects have not previously been taken into account.

The explanation is given in the introduction: “The main goal of the present work is to elaborate a new fabrication strategy to grow Ag-modified ZnO films for photocatalytic removal of paracetamol. In our recent work [35, 36], we proposed a synthetic method to prepare Ag/ZnO films via photo-fixation of Ag onto sol-gel derived ZnO substrate and the photocatalytic performance of the films have been investigated in the removal of Methylene blue and Malachite green dyes under UV and visible light illumination. Here, we extend our efforts by exploring the possibility to construct photocatalytic films from Ag-modified commercial ZnO powder as a simple and low-cost preparation method. Moreover, the degradation of PCA over pristine and Ag-modified ZnO films in distilled and drinking water under ultraviolet and direct sun-light was evaluated for the first time. It was found that under UV and natural sunlight in both types of waters, the modified ZnO catalysts with various concentrations of silver ions demonstrated increased drug mineralization capabilities compared to pure ZnO. Finally, the possible photocatalytic mechanism of PCA degradation over Ag-modified ZnO films is discussed.”

We described the experimental details for photo-fixation of silver ions in the text: “Chemical photodeposition was used to create silver co-catalytically modified ZnO films. After being immersed in aqueous silver nitrate solution for 20 min, the Ag/ZnO films were photo-fixed (irradiated) with UV illumination and then washed with water. The modified films were then dried at 100oC for 10 minutes to remove nitrate ions. In our previous research, we have successfully used the chemical photodeposition technique to photofix sol-gel films with silver ions and degrade organic dyes [35, 36].”

We have highlighted what is new in this research and what we have previously explored in our articles (in the abstract, introduction, and materials and methods)

  1. How can be the authors be sure that intermediate oxidation subproducts are not formed?

We are sure that no intermediates were formed during the photocatalytic processes because no new absorption peaks are present in the UV-vis spectra of the solutions. We prove this statement with the new figure - Figure 11. Absorbance spectra of the degradation of Paracetamol under natural sunlight (drug in drinking (a) and distilled (b) water) and ultraviolet (drug in drinking (c) and distilled (d) water)) illumination using Ag,10-2/ZnO powder film.  

The explanation for the figure is given in the text: “Finally, Figure 11 displays UV-vis spectra of paracetamol degradation in the pres-ence of Ag,10-2/ZnO photocatalyst, as obtained under natural sunlight and UV light in drinking and dis-tilled water. As previously reported [55], the spectrum of paracetamol shows bands at 194 and 243 nm, due to the π → π* and to the n → π* electronic transi-tions of the aromatic ring and the C=O group, respectively [55]. As can be seen, these bands gradually decline with irradiation time, in-dicating the efficient photocatalytic re-actions. According to the literature, photocatalytic degradation of paracetamol leads to the formation of non-toxic carboxylic acids as final products of the process. Since no ad-ditional bands appear in the spectra after the photocatalytic reactions, one may conclude that the presence of toxic by-products in the final solutions should be ruled out.”

  1. What is the final toxicity of the treated samples?

We would like to thank the reviewer for the question.  

The purpose of these experiments was to investigate the optimal concentration of the co-catalyst, the type of illumination (ultraviolet and natural sunlight) and waters (distilled and drinking) to achieve enhancement of the photocatalytic degradation of Paracetamol. Follow-up works, investigating the toxicity of the treated samples will be carried promptly. This is an investigation in our future experiments.

Reviewer 3 Report

In this article D. Ivanova et.al demonstrated the study of photocatalytic activity based on Ag/ZnO composite under Uv and natural solar light for the degradation of paracetamol in drinking and distilled water. The manuscript was well written and the concept theme is good which was using the Ag/ZnO nanoparticles. The results presented in the study are good and can be likely to attract the readers in in the field of photocatalytic applications. However, there are some issues that needs to be clarified before processing for the further step toward publication, and here are some of my concerns as given below.

1.      Abstract is too long. We understand the authors intention about their better results of their work. However, it could be better to the readers if authors could reduce the abstract length (in words around 25 will be in my opinion).

2.      In page No: 2-line No:52, the sentence He is much more should be modifying and rewrite, since the discussion is an object however it is rigid. So please treat the chemical as things or materials. English in the entire manuscript should further be polished.

3.      What is the novelty of your study as compared to your previous study (Catalysts 2023, 13(7), 1121;). Because similar type of Ag/ZnO based photocatalysts have been used for degradation works.Why does the author choose the ZnO nanostructure only for this research? Please emphasize the novelty of your structure. Also, please make a statement of choosing ZnO nanostructure in your research as compared to another nanostructure. Introduction: (1) Why study ZnO based nanoparticles their wide application in various filed and how does it helpful to enhance photocatalytic properties. Some of the latest review reports are suggested to cite at the MXene related discussions, such as Materials Science and Engineering: B 289 (2023): 116263. ACS Materials Letters 5 (2023): 2739-2746. Materials Today Communications (2023): 106840. Angewandte Chemie 135.23 (2023): e202304301., Journal of Alloys and Compounds (2023): 170841.

4.      If the author uses any abbreviation words as first time please use the full form for example: PEG 4000.

5.      Authors should clearly describe the experimental details and formation of Ag, 10-2/ZnO composite formation.

6.      SEM images are not clear to observe the structural morphology. Please provide the zoomed view of your materials.

7.      Please change the Figure 8, 9 color for better look, and better understanding by distinguish the differences in the Kinetics of removal of paracetamol in distilled.

8.      Please provide the band analysis of the as-prepared Ag, 10-2/ZnO powder films for better understanding to the readers.

9.      Since Ag, 10-2/ZnO is a metal oxide-based material, in which O 1s cal influence the optoelectrical properties of the device. Please explain it in the revised manuscript. How about the stability of your photocatalyst?

There are some sentence grammatical errors and some parts of the manuscript needs the English language corrections. 

Author Response

Response to Reviewer 3

Ms. Ref. No.: 2646932

Title: “Degradation of Paracetamol in Distilled and Drinking Water via Ag/ZnO Photocatalysis Under UV and Natural Sunlight”, written by Dobrina Ivanova, George Tzvetkov, Nina Kaneva

            Water

We thank very much to the Reviewer for the valuable marks, comments and suggestions, supporting the presentation of our work. We accept all the recommendations and corrected our paper properly following the suggestions given by the Reviewer in the same sequence.

Hope you find revised manuscript suitable for publication in Water. We look forward to hearing from you in due course.

Reviewer 3: In this article D. Ivanova et.al demonstrated the study of photocatalytic activity based on Ag/ZnO composite under Uv and natural solar light for the degradation of paracetamol in drinking and distilled water. The manuscript was well written and the concept theme is good which was using the Ag/ZnO nanoparticles. The results presented in the study are good and can be likely to attract the readers in in the field of photocatalytic applications. However, there are some issues that needs to be clarified before processing for the further step toward publication, and here are some of my concerns as given below:

  1. Abstract is too long. We understand the authors intention about their better results of their work. However, it could be better to the readers if authors could reduce the abstract length (in words around 25 will be in my opinion).

We reduced the abstract length, as recommended. The new reduced abstract is: “The present study demonstrates the synthesis and application of Ag/ZnO powder films (thickness of 4 μm) as photocatalysts for natural sunlight and ultraviolet (UV, 315–400 nm) irradiation. The synthesis procedure is simple and eco-friendly, based on the photo-fixation of silver ions onto commercial ZnO powder via UV illumination for the first time. The photocatalytic efficiency of the newly developed films is evaluated through degradation of paracetamol in distilled and drinking water. Our experimental evidences showed that the Ag/ZnO nanostructure films are much more active than pristine ZnO films in the photodegradation process. Namely, the photocatalytic efficiency of the films modified with 10-2 M concentration of silver ions achieved the highest degradation (D) percentages for paracetamol in both types of waters (Ddistilled = 80.97%, Ddrinking = 82.5%) at natural sunlight. Under UV exposure, the degradation percentages are slightly lower but still higher than those achieved by pure ZnO films (Ddistilled = 53.13%, Ddrinking = 61.87%). It has been found that the photocatalytic activity grows in direct proportion to the concentration of Ag+ ions: ZnO < Ag,10-4/ZnO, < Ag,10-3/ZnO < Ag,10-2/ZnO. Scanning electron microscopy, X-ray diffraction, X-ray photoelectron spectroscopy, UV-vis diffuse reflectance and photoluminescence spectroscopy are used to characterize the as-prepared ZnO and Ag/ZnO nanostructures. The improved photocatalytic performance of the Ag/ZnO films is mostly attributed to the combination of excited electron transfer from ZnO to Ag and the inhibition of photogenerated electron-hole pair recombination. Furthermore, Ag/ZnO nanostructure films could retain their photocatalytic activity even after three cycles of use, highlighting their potential practical application for the treatment of pharmaceutical wastewater in real-world scenarios, where natural sunlight is often more readily available than artificial UV light.”

  1. In page No: 2-line No:52, the sentence He is much more should be modifying and rewrite, since the discussion is an object however it is rigid. So please treat the chemical as things or materials. English in the entire manuscript should further be polished.

We completely agree with the reviewer, we deleted the sentence:  “He is much more harmful when taken in excess [11, 12] than other painkillers. Hepatotoxicity, skin, liver, and kidney damage are some of its negative effects at high doses, and it can also alter how some medications (rifampicin, cimetidine, chloramphenicol, and busulfan) interact with one another.”

The English language throughout the article has been corrected.

  1. What is the novelty of your study as compared to your previous study (Catalysts 2023, 13(7), 1121;). Because similar type of Ag/ZnO based photocatalysts have been used for degradation works. Why does the author choose the ZnO nanostructure only for this research? Please emphasize the novelty of your structure.

In our previous two studies - Ivanova, D.; Mladenova, R.; Kolev, H.; Kaneva, N. Effect of Ultraviolet Illumination on the Fixation of Silver Ions on Zinc Oxide Films and their Photocatalytic Efficiency. Catalysts 2023, 13, 1121.; Kaneva, N.; Bojinova, A.; Papazova, K. Enhanced Removal of Organic Dyes Using Co-Catalytic Ag-Modified ZnO and TiO2 Sol-Gel Photocatalysts, Catalysts 2023, 13, 245., we synthesize sol-gel thin films (ZnO, TiO2) using the following reagents – 2-methoxyethanol, monoethanolamine, zinc acetate dehydrate, titanium isopropoxide. The method of photofixation and the method of modification with silver ions is the same. In these two papers, we degrade the organic dyes Malachite Green and Methylene Blue in the presence of ultraviolet, visible light and in the dark.

In the present paper, we modify with silver ions films prepared from commercial ZnO powder for the first time. We believe that this fabrication strategy is much faster and cost-effective. The use of sunlight to break down paracetamol in drinking and distilled water is also a novelty in this study. All these effects have not previously been taken into account.

The explanation is given in the introduction: “The main goal of the present work is to elaborate a new fabrication strategy to grow Ag-modified ZnO films for photocatalytic removal of paracetamol. In our recent work [35, 36], we proposed a synthetic method to prepare Ag/ZnO films via photo-fixation of Ag onto sol-gel derived ZnO substrate and the photocatalytic performance of the films have been investigated in the removal of Methylene blue and Malachite green dyes under UV and visible light illumination. Here, we extend our efforts by exploring the possibility to construct photocatalytic films from Ag-modified commercial ZnO powder as a simple and low-cost preparation method. Moreover, the degradation of PCA over pristine and Ag-modified ZnO films in distilled and drinking water under ultraviolet and direct sun-light was evaluated for the first time. It was found that under UV and natural sunlight in both types of waters, the modified ZnO catalysts with various concentrations of silver ions demonstrated increased drug mineralization capabilities compared to pure ZnO. Finally, the possible photocatalytic mechanism of PCA degradation over Ag-modified ZnO films is discussed.”

Also, please make a statement of choosing ZnO nanostructure in your research as compared to another nanostructure. Introduction: (1) Why study ZnO based nanoparticles their wide application in various filed and how does it helpful to enhance photocatalytic properties.

We thank the referee for this remark. Through these comments, we have modified the introduction and thus made it more extensive: “Heterogeneous photocatalysis with semiconductor materials like zinc oxide (ZnO) has indeed gained popularity as an effective method for treating wastewater contami-nated with dyes, pesticides, and pharmaceuticals [10]. ZnO is a low-cost, non-toxic, chemically stable and easy to produce material with a range of applications, including micro-gas sensing devices [19], solar cells [20], nonlinear optics [21], integrated photonic devices [22] and photocatalysts for wastewater treatment [23, 24]. However, under visible or sunlight irradiation ZnO cannot be used as a catalyst due to its significant band gap energy. Additionally, due to the rapid recombination rate of charge (e-/h+) couples, its photocatalytic efficiency is also constrained. To overcome this problem, ZnO surface can be modified and functionalized with co-catalysts, typically noble metals as Pt, Pd, Ag, and Au, and based on the formation of heterojunctions to limit the possibilities for re-combination [25–27]. Different approaches have been used to combine the ZnO substrate with metallic co-catalysts [25-27]. Among them, the direct photofixation or photocatali-tycally deposition of metal clusters under UV light gained attention recently [28, 29]. More precisely, under the influence of the photogenerated charges on the ZnO substrate, the co-catalyst can be deposited as a result of the reduction of metal ions, which are typically from an aqueous solution [28, 29]. As photogenerated electron acceptors, the noble metal ions (Ag+, Pt2+, Pd2+, Au3+) can be successfully reduced to the corresponding metals [30-32].

Most of the studies that have examined ZnO and co-catalytically modified ZnO focus on the systems using suspended photocatalyst particles. This limits their practical applicability due to the requirement of centrifugation or filtration to reuse fine ZnO par-ticles. The design of systems where the photocatalysts are supported on inert substrates as layers and thin films proved to be successful in overcoming such obstacles [30]. Very recently, Hao and co-workers [33] described the construction of hydrothermally grown ZnO films on wire mesh, subsequently decorated with Ag nanospheres using the im-pregnating photoreduction treatment. The formed Ag–ZnO heterojunction improves the absorption of UV and visible light, thereby boosting the photocatalytic properties of the films. Rati and co-workers [34] reported that Ag addition contributes to the improvement of ZnO thin films throughout the photocatalytic process of Methylene blue degradation. Therefore, developing novel ZnO-based thin film photocatalysts for water remediation is of constant interest.”

Some of the latest review reports are suggested to cite at the MXene related discussions, such as Materials Science and Engineering: B 289 (2023): 116263. ACS Materials Letters 5 (2023): 2739-2746. Materials Today Communications (2023): 106840. Angewandte Chemie 135.23 (2023): e202304301., Journal of Alloys and Compounds (2023): 170841..

We added these reports in the introduction. In the References, these sources are marked in blue.

  1. If the author uses any abbreviation words as first time please use the full form for example: PEG 4000.

We added 4000 to the name polyethyleneglycol in the section 2.1.Materials (Materials and Methods): “Zinc oxide commercial powder (≥ 99.0 %), polyethyleneglycol 4000 (PEG 4000), ethanol (C2H5OH, ≥ 99.0 %) and….”. And then everywhere in the text the abbreviated name is used PEG4000.

  1. Authors should clearly describe the experimental details and formation of Ag, 10-2/ZnO composite formation.

We described the experimental details for photo-fixation of silver ions in the text: “Chemical photodeposition was used to create silver co-catalytically modified ZnO films. After being immersed in aqueous silver nitrate solution for 20 min, the Ag/ZnO films were photo-fixed (irradiated) with UV illumination and then washed with water. The modified films were then dried at 100oC for 10 minutes to remove nitrate ions. In our previous research, we have successfully used the chemical photodeposition technique to photofix sol-gel films with silver ions and degrade organic dyes [35, 36].”

We have highlighted what is new in this research and what we have previously explored in our articles.

  1. SEM images are not clear to observe the structural morphology. Please provide the zoomed view of your materials.

We provided an insert of SEM images (Figure 3a and Figure 3b) with higher magnifications.

  1. Please change the Figure 8, 9 color for better look, and better understanding by distinguish the differences in the Kinetics of removal of paracetamol in distilled.

We changed Figure 8, 9, as recommended. The Figures are presented in color.

  1. Please provide the band analysis of the as-prepared Ag, 10-2/ZnO powder films for better understanding to the readers.

According the referee’s remark, a new Figure is included in the revised manuscript, showing a scheme of the photocatalytic process over the film.

  1. Since Ag, 10-2/ZnO is a metal oxide-based material, in which O 1s cal influence the optoelectrical properties of the device. Please explain it in the revised manuscript. How about the stability of your photocatalyst?

We thank the referee for this remark. In order to fully describe the properties of the 10-2/ZnO film, the deconvoluted O 1s XPS spectra of ZnO and 10-2/ZnO are compared (Figure 5c) and discussed in the revised manuscript. As can be seen, the spectral lineshapes and the resulting components are practically identical. Thus, one can conclude that the oxygen environment for ZnO and Ag, 10-2/ZnO films is very similar.

“Figure 5c shows the details of the O 1s peaks deconvoluted using Gaussian lineshapes. Each peak is deconvoluted into three components at 530.7 eV (OI), 532.6 eV (OII), and 534.1 eV (OIII) corresponding to O2- ions in ZnO, oxygen vacancies and/or OH groups, and physisorbed water molecules [41]. As seen, the components for both films show equal intensity proportions. This result suggests that the oxygen environment for ZnO and Ag, 10-2/ZnO films is very similar. Interestingly, the HR scan in the 380-360 eV BE range for Ag, 10-2/ZnO film (Figure 5d) detects two weak signals at 368.2 eV and 374.8 eV. These peaks can be attributed to the Ag 3d5/2 and Ag 3d3/2, supporting the presence of metallic Ag [42]. Contrary, XPS examination of ZnO film does not show the existence of Ag.”

Furthermore, SEM image of the spent Ag modified ZnO film is also included. The image reveals that the integrity and surface morphology is retained, thus highlighting its stability after the cyclic experiment.   

Round 2

Reviewer 2 Report

The manuscript has been improved and it can be accepted for publication.

Minor grammar final checking maybe necessary